# On the Evolution of Different Types of Green Water Events

**Jassiel V. H. Fontes** [1], **Irving D. Hernández** [2], **Edgar Mendoza** [3,*], **Rodolfo Silva** [3], **Eliana Brandão da Silva** [1], **Matheus Rocha de Sousa** [1], **José Gonzaga** [1], **Raíssa S. F. Kamezaki** [1], **Lizeth Torres** [3] and **Paulo T. T. Esperança** [4]

[1] Departamento de Engenharia Naval, Escola Superior de Tecnologia, Universidade do Estado do Amazonas, Manaus 69050-020, Brazil; jvfontes@uea.edu.br (J.V.H.F.); ebs.eng17@uea.edu.br (E.B.d.S.); mrs.gen18@uea.edu.br (M.R.d.S.); jgsn.eng17@uea.edu.br (J.G.); rsfk.gen18@uea.edu.br (R.S.F.K.)

[2] Núcleo de Estruturas Oceânicas—NEO, Programa de Engenharia Oceânica, COPPE, Universidade Federal do Rio de Janeiro, Rio de Janeiro 20945-970, Brazil; irving.david@coppe.ufrj.br

[3] Instituto de Ingeniería, Universidad Nacional Autónoma de México, Mexico City 04510, Mexico; rsilvac@iingen.unam.mx (R.S.); ftorreso@iingen.unam.mx (L.T.)

[4] Laboratório de Tecnologia Oceânica—LabOceano, Programa de Engenharia Oceânica, COPPE, Universidade Federal do Rio de Janeiro, Rio de Janeiro 21941-907, Brazil; ptarso@laboceano.coppe.ufrj.br

* Correspondence: emendozab@iingen.unam.mx

**Abstract:** Green water events may present different features in the initial stage of interaction with the deck of a structure. It is therefore important to investigate the evolution of different types of green water, since not all the events interact with the deck at the same time. In this paper, the evolution of three types of green water events (dam-break, plunging-dam-break, and hammer-fist) are studied. The water surface elevations and volumes over the deck in consecutive green water events, generated by incident [wave trains in a wave flume, were analyzed using image-based methods. The results show multiple-valued water surface elevations in the early stage of plunging-dam-break and hammer-fist type events. Detailed experimental measurements of this stage are shown for the first time. The effect of wave steepness on the duration of the events, maximum freeboard exceedance, and volumes were also investigated. Although the hammer-fist type showed high freeboard exceedances, the plunging-dam-break type presented the largest volumes over the deck. Some challenges for further assessments of green water propagation are reported.

**Keywords:** green water events; multiple surfaces; wave trains; image analysis; wave flume; volumes; challenges; freeboard exceedance; plunging-dam-break; hammer-fist

## 1. Introduction

In ocean and offshore engineering, the phenomenon in which incident waves overtop structures and propagate over their decks is known as green water [1–3]. This water invasion may affect the operation of marine structures and vessels. The water that propagates over the deck or impacts the installations located on it, including crew checkpoints, may also be sufficiently dangerous to cause injury to humans. This water can also damage delicate equipment or structures installed on the deck. When the structures are floating, the additional water mass over the deck can alter the dynamics and endanger the integrity of the structure [1,2,4].

In ocean engineering research, experimental observations have suggested various patterns used to characterize different types of green water. Broken flow features are often seen, if they are generated by breaking waves [5–8], or air cavity formation at the start of the deck, if the incident waves are unbroken or partially broken [9–14]. In 2007, a classification of different types of green water was suggested by [9], who described the different events found in regular wave trains: The dam-break (DB), the plunging-dam-break (PDB), and the hammer-fist (HF) types. These events are mainly differentiated by the features formed by the flow at the start of the deck, and have motivated further green water research [11,12,15–18].

The main difference between the DB and the PDB types of green water is that an air cavity is often formed at the deck's edge in the latter. As water propagates over the deck, this cavity is entrapped and fragments inside the flow, producing bubbles that tend to escape when the water travels along the deck [19]. In the HF type, no cavities are formed; instead, a block of water remains almost suspended at the beginning of the deck, as a fluid-arm [9]. Then, as the incident wave advances over the deck, this arm falls, and hits the deck (see details in [9,12]). Improving the knowledge of these events' physical behavior, including their formation at the beginning of the deck and their evolution is relevant. It allows more precise comparisons and validation of numerical models. Engineers can use these numerical tools to predict the water propagation over the decks of structures and estimate their induced loads. In turn, the design methods could benefit from better predictions minimizing green water occurrence and adding sufficient strength to resist it. Some studies that investigate the propagation of water over the deck of a marine structure have used obstructive wave probe sensors to obtain information on flow evolution based on the variation of water surface elevations; for instance, see [1,13,20–23]. It is known that these types of sensors, which can be conductive [24], capacitive [25], or resistive [26], provide time series of single-valued measurements of water elevation at a specific location [27] (see single-valued water surface elevation, SVWS, in Figure 1a). Nevertheless, for some applications, such as in overturning surfaces, conventional sensors cannot simultaneously measure multiple surfaces [28]. Yao and Hu [28] explained that some phenomena could have triple-valued water surface amplitudes, such as the air trapped beneath an overturning plunger. Furthermore, the snapshots presented in [9,12] clearly show that the initial stages of some shipping water events have these features.

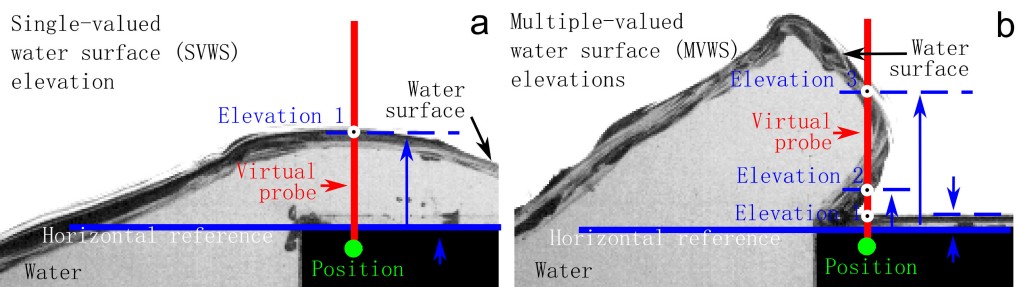

**Figure 1.** Definition of water surface measurements at a specific location over a horizontal domain. (**a**) Single-valued water surface (SVWS) elevation. (**b**) Multiple-valued water surface (MVWS) elevations. Images were modified from the database provided by [12].

Recently, Hernández-Fontes et al. [29] introduced the concept of multiple-valued water surface (MVWS) elevations to refer to the amplitudes of the several water surfaces that can be measured at a specific location, along a horizontal domain, in some 2D hydrodynamic experiments, as illustrated in Figure 1b. Hereafter, the SVWS and MVWS definitions are employed to describe this work.

Regarding the research of green water evolution, in terms of water elevations, recent investigations have been performed using image-based measurements in simplified experiments, providing useful information [17,19,30,31]. From these works, Hernández-Fontes et al. [31] employed the concept of Virtual Wave Probes (VWP, [32]) in green water research to investigate the distribution of water-on-deck through binary image processing and analysis. This simplified image-based approach allows a time series of effective water heights at the desired positions over the deck of the structure to be obtained. As the method allows the spatio-temporal evolution of green water heights to be measured, it is useful to obtain experimental data for validating analytical approaches [17,30]. In [30], isolated green water events, generated by the wet dam-break approach, were considered to validate the applicability of an analytical convolution model to capture the flow evolution. Three study cases were defined, varying the freeboard of the structure and considering incident wet dam-break bores. The green water features recorded gave types, which were basically DB

and PDB with small cavities. Comparisons with SVWS elevation measurements obtained with the method proposed by [31] were possible. The work of [30] was recently extended by [17,33,34] to evaluate the water elevations in consecutive green water events generated by incident wave trains. In those works, the use of the image-based method of [31] was used to obtain SVWS measurements from the experiments. However, in the experiments of those works (see [17]), it was reported that with steep waves, events resembling HF types were produced. Therefore, new approaches to investigate the water elevations should be considered, because this type of event presents MVWS elevations. To contribute to this, Hernández-Fontes et al. [29] extended the method of [31], providing an open-source and simplified image-based approach to obtain MVWS elevation measurements, as shown in Figure 1b.

Recently, Hernández-Fontes et al. [12] provided a detailed investigation of some of the features of different types of green water events generated by incoming wave trains in wave flume experiments. With the use of a high-speed camera, the main visual features of the consecutive green water events were discussed, providing, for the first time, a video database with the details of the events, captured at 250 fps. Details of the evolution of different types of green water events were reported. In that work, a classification of different types of green water was presented, suggesting the use of "apparent" and "effective" interaction parameters to consider green water events. However, analysis of other parameters is still required to provide further knowledge concerning green water events on rectangular-shaped structures, such as the MVWS elevations observed in the generation of some events, the volumes over the deck, and the freeboard exceedances. The latter corresponds to the water elevations found at the start of the deck and are relevant data for the implementation of analytical and numerical models. To contribute to this gap, this paper extends the work of [12] to investigate the evolution of different types of green water events generated with incident wave trains in wave flume experiments. Using image-based approaches, the water surface elevations and volumes over the deck were analyzed.

- The spatiotemporal distribution of SVWS and MVWS elevations and volumes found in different types of green water events was analyzed. Experimental measurements of water surface elevations during the formation of different kinds of green water at the start of the deck are presented for the first time. Providing knowledge on the evolution of green water events, from their generation at the beginning of the deck and their propagation on it, can help to perform more detailed comparisons of analytical and numerical results. To the best of the authors' knowledge, model comparisons have been made using SVWS elevations until now.
- The effect of wave steepness on green water duration, maximum freeboard exceedance, and maximum volumes over the deck was analyzed. The study of these parameters can be critical for differentiating the behavior of different types of green water events. Research on the differences found in the maximum freeboard exceedance in PDB and HF type events is still scarce. The study of volumes over the deck can indicate the expected vertical loading for these types of events.
- A discussion of challenges to assess the evolution of different types of green water events is included in this paper, which could help further green water research.

The manuscript is organized as follows: Section 2 presents the experimental methods, describing the experimental data and the image-based methods employed for the analysis. The green water surface elevations and volumes, including the effects of wave steepness on them, are shown in Sections 3 and 4, respectively. Sections 5 and 6 discuss the challenges to assess green water propagation and main conclusions, respectively.

## 2. Experimental Methods

### 2.1. The Green Water Experiments

In this study, experiments of consecutive green water events on a fixed structure, caused by incoming wave trains, in a wave flume, were considered. Figure 2 shows the

side and top views of the experimental setup, which consists of a rectangular structure installed inside a 22 m length wave flume. The flume contains a piston-type wavemaker that generated the incident regular wave trains, which in turn produced green water events on the deck of the structure when the initial freeboard of 0.05 m was exceeded. A high-speed camera model, Fastec Imaging HiSpec2GMono (Fastec Imaging, San Diego, CA, United States) [35], with an additional 50 mm lens, was used to capture the events at 250 fps. This study extends the work of [12], using the video database of that work to perform the image-based measurements. The reader is referred to [12] for details of the repeatability of the experiments and the qualitative classification of the green water events obtained. The time scale in the results considered in this work corresponds to the video frames (i.e., the first frame of each video is the initial time). Information of the time with respect to the wavemaker is found in the database [12].

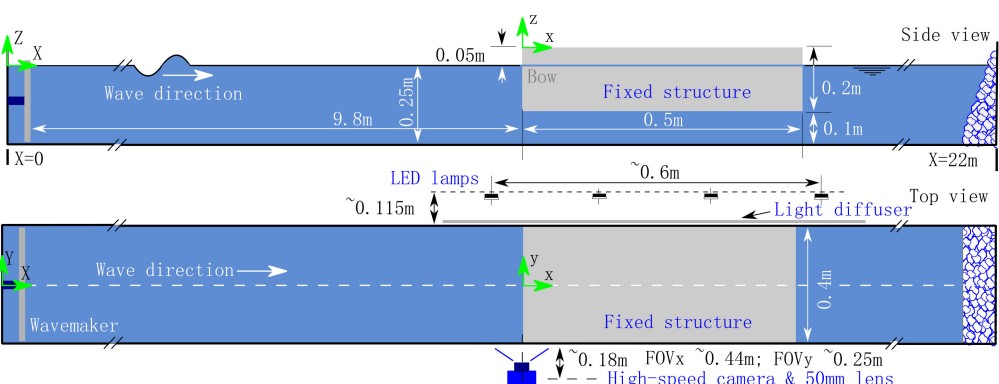

**Figure 2.** (**Side**) and (**top**) views of the experimental setup.

Table 1 shows the fifteen study cases considered in this work, including the design parameters of the regular waves used to activate the wavemaker into generating the wave trains. It is important to mention that the features of some regular waves, particularly the steepest ones, may have been influenced by the working depth of the flume (see [12]). Each train contained more than fifteen waves, with most caused green water events. However, for a systematic evaluation, only the first eight events in each study case were considered, as in [12]. Note that the letter of the case name represents a wave length (CA-CE, 2.5–0.75 m), while its number represents a wave height (1–3, 0.08–0.16 m). The wave trains generated different types of green water (see the classification of the events in [12]). PDB and HF types formed MVWS at the bow, as shown in Figure 1b, whereas the DB-types formed SVWS (Figure 1a). In some PDB types, very small cavities were formed at the bow; however, small 3D effects in the flow did not allow differentiation of the air from the water (see the video database provided by [12]).

### 2.2. The Image-Based Approaches

In this study, image-based methods were employed to take measurements from the available video data. The main parameters to be measured were the time series of water surface elevations and the water volumes over the deck.

### 2.2.1. MVWS Elevation Measurements

To perform the MVWS elevation measurements, the gray-scale videos in the database (resolution of 769 × 441 px) were binarized in order to perform morphological operations, following the procedures presented in [29,31], for use in ImageJ software (NIH, Bethesda, MD, USA). These works offered simplified procedures and open-source software to obtain time series of effective water heights [31] and SVWS and MVWS elevation measurements [29] in 2D hydrodynamic experiments, disregarding the differentiation of possible 3D effects on the water surface and bubbles entrapped in the flow.

**Table 1.** Design regular wave parameters sent to the wavemaker to generate the incident wave trains [12].

| Case Name | Length | Height |
|:---:|:---:|:---:|
| | $(L_w, \text{m})$ | $(H_w, \text{m})$ [1] |
| CA1 | | 0.08 |
| CA2 | 2.50 | 0.12 |
| CA3 | | 0.16 |
| CB1 | | 0.08 |
| CB2 | 2.00 | 0.12 |
| CB3 | | 0.16 |
| CC1 | | 0.08 |
| CC2 | 1.50 | 0.12 |
| CC3 | | 0.16 |
| CD1 | | 0.08 |
| CD2 | 1.00 | 0.12 |
| CD3 | | 0.16 |
| CE1 | | 0.08 |
| CE2 | 0.75 | 0.12 |
| CE3 | | 0.16 |

[1] Wave heights 0.08 m, 0.12 m, and 0.16 m are referred to in this work as $H_{w1}$, $H_{w2}$, and $H_{w3}$, respectively.

In this work, the procedure to obtain the measurements of MVWS elevations based on contours [29] was adapted for the present configuration of the input videos. There are two main stages in this method: Image processing and analysis, which are available in ImageJ plugins for free use [29]. Figure 3 illustrates the procedure to obtain the measurements for the data used in the present work:

Image processing: The objective of this stage is to obtain binarized images from the input images, which are in gray-scale, with a resolution of 769 × 441 px. This is done through changes in the Window and Level parameters (428 and 63, respectively) and the selection of threshold values according to the user criteria (Otsu method; values: 147, 255, Figure 3). See Appendix A for the possible error associated to this selection. Once the images are binarized, dilation and erosion procedures allow contours of flow surfaces to be obtained.

Image analysis: The objective of this stage is to perform the water surface measurements from the motion of the contours defined inside the regions of interest (vertical rectangles of unitary width, or virtual wave probes, VWPs) located over the horizontal domain of the images. Inside these regions, the pixel variations are converted to surface elevation measurements.

For the purposes of the present work, several VWPs were distributed along the horizontal domain of the images to obtain water surface elevations of the incident wave and the water on deck, denoted by "*w*" and "*d*", respectively, shown in Figure 4. Note that the spacing between the probes differs for the different regions in the image. The number next to "*w*" and "*d*" indicates the distance, in mm, with respect to the bow edge ($x = 0$). Some VWPs (*w2*, *d2-d58*) were spaced every 2 mm, to capture details of the flow during the formation of the green water events at the bow. Note the probes at every 10 mm to capture the water on deck and the incident wave (*d10–d90* and *w10–w80*, respectively).

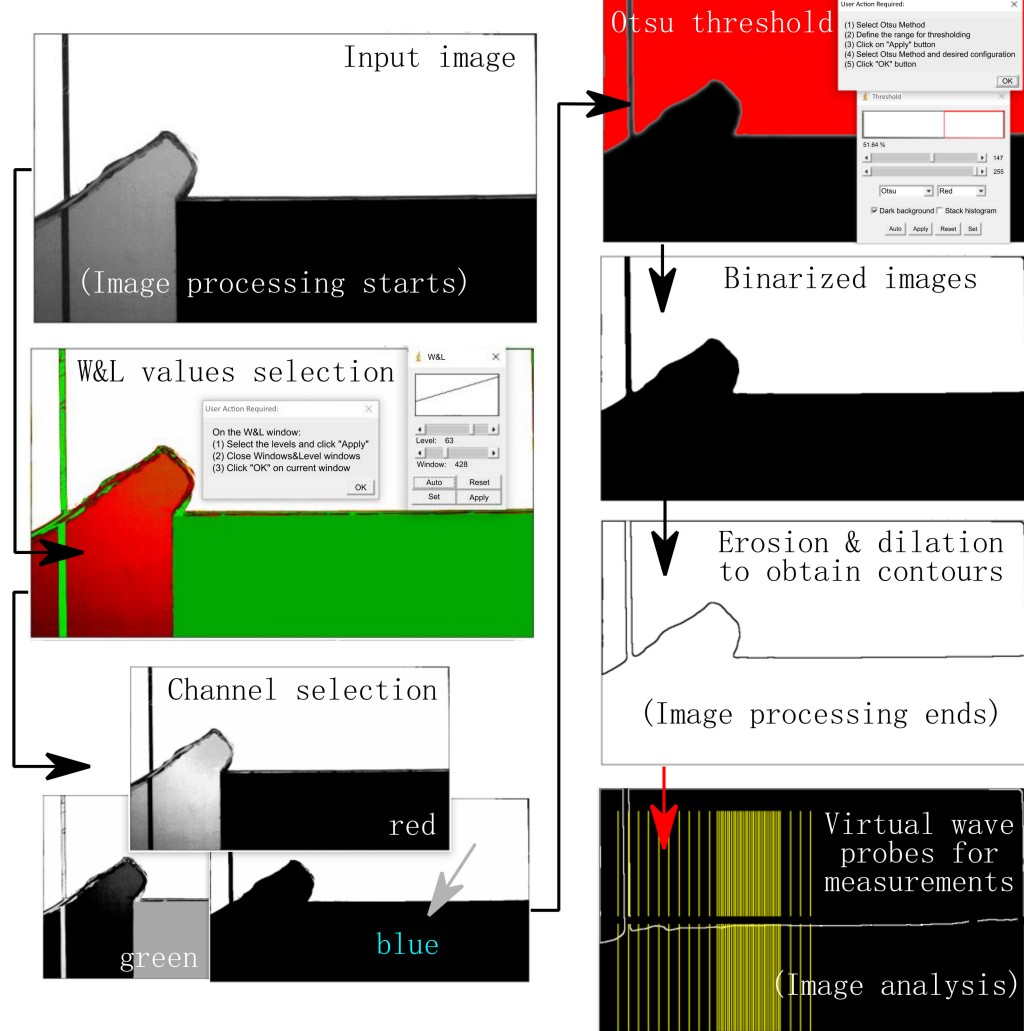

**Figure 3.** Workflow during the image processing and analysis stages of this work, used to obtain the water surface elevations.

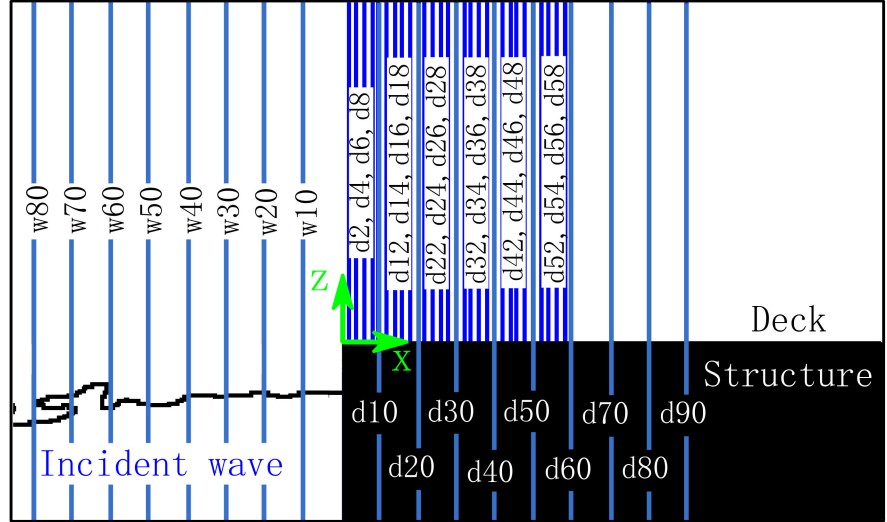

**Figure 4.** Distribution of VWPs along the horizontal domain of the images. The letters "*d*" and "*w*" mean "over the deck (+*x*)" and "wave, outside the deck (−*x*)", respectively. The number next to these letters indicates the distance, in mm, from the origin (*x* = 0).

### 2.2.2. Green Water Volumes

To measure the volume of water over the deck, a region of interest (rectangle) was defined over the deck, in the binarized images (images without the erosion and dilation procedures, Figure 3), as shown in Figure 5. Inside this area, the time series of total area of white pixels (or black, if the image colors are inverted), representing the water over the deck, were measured. This area represented the volume of water over the deck, disregarding 3D effects on the surface and considering a constant width of 0.4 m (structure width). Hydrostatic vertical loads ($F_h$) can be assumed with these volumes: $F_h = \rho g \nabla$, where $\rho$ is density (in kg/m$^3$), $g$ is acceleration due to gravity (in m/s$^2$), and $\nabla$ is water volume (in m$^3$).

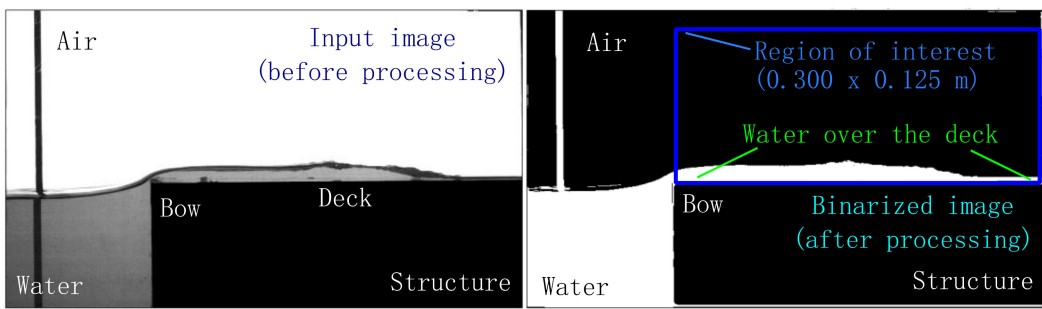

**Figure 5.** Definition of the region of interest to measure the green water volume over the deck. (**Left**): Input image; (**right**): Binarized image with the region of interest in which the volumes (white area in the figure) were measured.

It is important to mention that the deck resembles a semi-infinite domain for propagation, in which there is no backflow due to reflection by a downstream wall (as in, for instance [36–41]); instead, the flow propagates entirely over the deck, falling into the flume at the end of the structure.

## 3. Green Water Surface Elevations

### 3.1. Evolution of the Different Types of Green Water Events

The measurements of water surface elevations ($\eta$) over the deck with the virtual sensors defined in Figure 4 allowed the visualization of the evolution of different types of green water events. This is perhaps the first time that details of the evolution of some events, such as HF, has been reported experimentally. Although videos of all the consecutive events of the fifteen study cases provided by [12] have been processed to obtain the measurements, only one event of the cases generated with $H_{w3}$ (i.e., CA3, CB3, CC3, CD3, and CE3) is described in this section. Figure 6a–e shows the temporal evolution of $\eta$ and some illustrative snapshots for each case.

Figure 6a shows the evolution (i.e., time series of water surface elevations, $\eta$) of the sixth event in CA3. This event can be considered as a DB type of green water since no significant cavity is formed. After image processing, it showed SVWS elevations, as seen in the time series of $\eta$. SVWS elevation measurements are the type of data usually considered in green water research using conventional wave gauges (e.g., [1,2,17,22]). However, in [9,12,13,15,16] and other works, it has been reported that other types of events present particular features at the initial stage of the interaction with the deck, which require the use of the MVWS elevation concept [29]. This can be observed in the evolution of the events shown in Figure 6b–e, which can be considered as PDB-type, a transition between PBD and HF types, HF-type and one HF case, with considerable 3D effects in the free surface, respectively. Unlike the curve trends shown in Figure 6a, these events have several $\eta$ values at each position over the first part of the deck. In all cases, the freeboard exceedance, quantified by the sensor *d2*, shows SVWS elevations. Then, before the green water interacts with the deck, double-valued (e.g., Figure 6b) or multiple-valued water surfaces (e.g., Figure 6c–e) are observed. Finally, when the interaction with the deck has

occurred, SVWS are observed again, and the flow behavior is like that seen in Figure 6a. It can be concluded that MVWS matters during the formation of the events and the initial interaction with the deck.

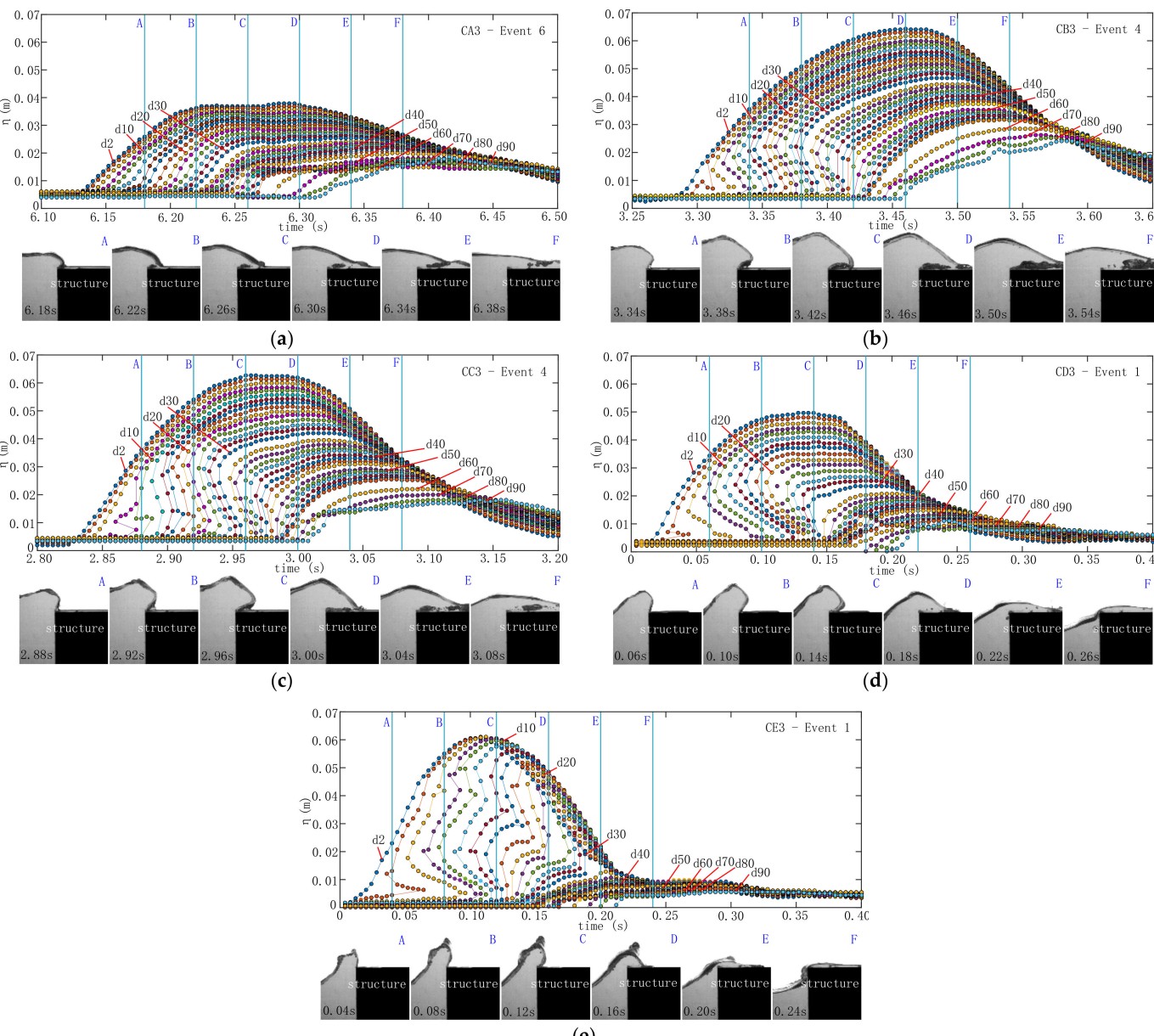

**Figure 6.** Evolution of different types of green water events. Above: Time series of water surface elevations ($\eta$) obtained by virtual probes *d2* to *d90*, according to Table 1. Below: Representative snapshots at different stages during the event (A–F, [12]). (**a**) Evolution of the sixth event of CA3. (**b**) Evolution of the fourth event of CB3. (**c**) Evolution of the fourth event of CC3. (**d**) Evolution of the first event of CD3. (**e**) Evolution of the first event of CE3.

### 3.2. The Effects of Wave Steepness on Green Water Parameters

In green water research, the time series of water elevation measured at the start of the deck are known as freeboard exceedance [1,20,42] and are important in the application of analytical and numerical models used in ocean engineering to analyze the water-on-deck behavior, such as [16,19,43]. In this research, the measurements given by sensor *d2* defined in Figure 4 (positive measurements from the deck level), are considered as the freeboard exceedance of the green water events. Figure 7 shows the evolution of the water elevation of the incident waves, at the deck level, measured by probes *w10* to *w80* ($x = -0.01$ m to

$x = -0.08$ m, Figure 4), for some representative cases. The crests of the incident waves mark the difference between the trend in the flow outside the deck and the flow at the start of the deck (freeboard exceedance at $x = 0.002$ m, probe *d2*).

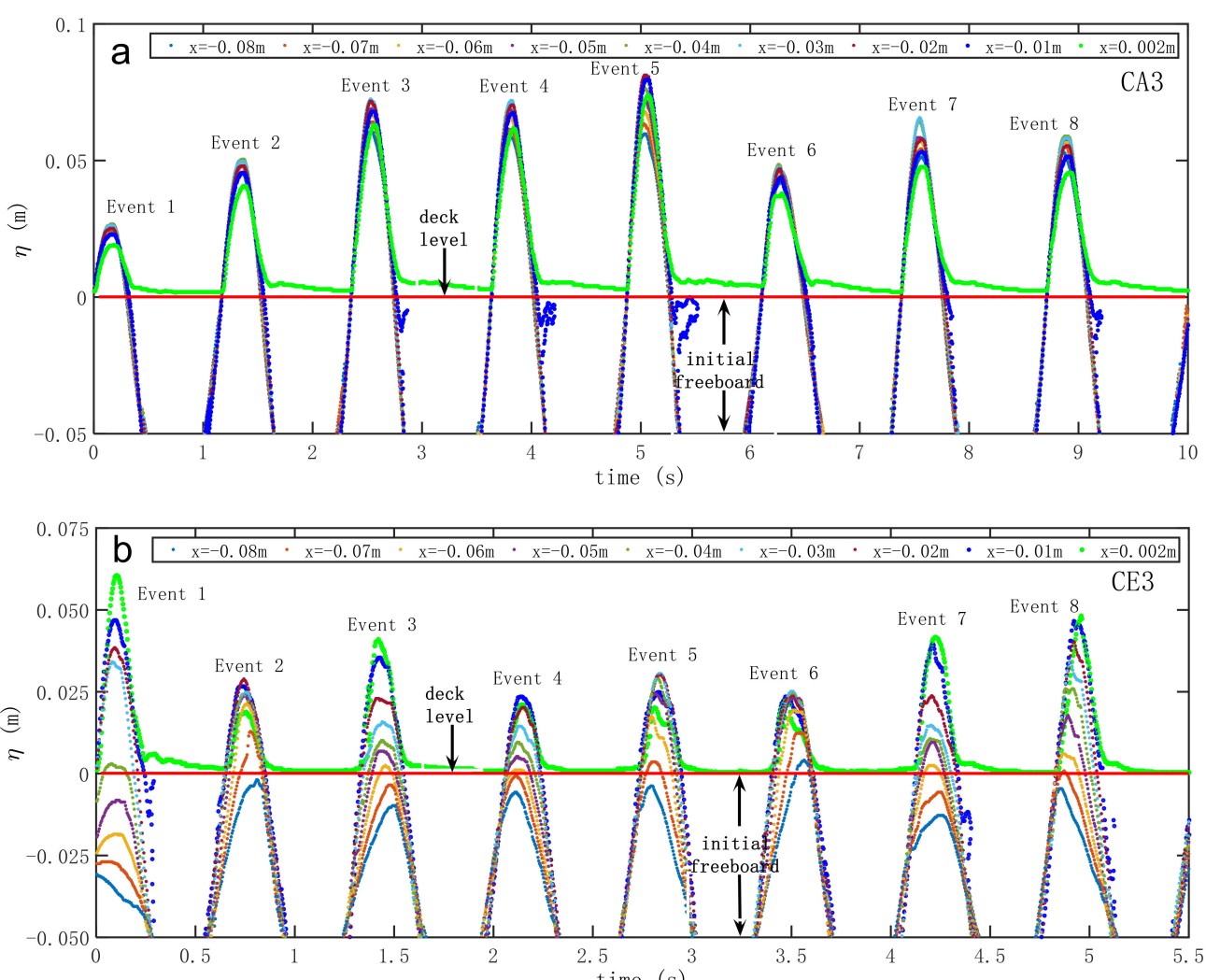

**Figure 7.** Time series of water surface elevations ($\eta$) obtained from virtual probe *d2* (over the deck, $x = 0.002$ m) and from probes *w10* to *w80* (outside the deck, from $x = -0.01$ m to $x = -0.08$ m), for eight consecutive green water events. (**a**) Data for CA3. (**b**) Data for CE3.

Figure 7a shows the results for a case obtained with long waves (CA3, $\varepsilon = 0.064$, where $\varepsilon$ is wave steepness, $H_w/L_w$). In this case, most of the green water events were of the PBD-type (see [12]). Note that in all the events, the freeboard exceedance is always lower than the wave elevation approaching the structure (e.g., probe *w10*, $x = -0.01$ m). However, this behavior is not observed for some events generated with steeper waves, as shown in Figure 7b (CE3, $\varepsilon = 0.21$). In HF-type events 1, 3, 7, and 8, shown in this figure, the freeboard exceedance is higher than the elevations of the wave close to the structure. This shows that for analytical or numerical approaches, which require incident wave information to predefine the freeboard exceedance, special consideration should be given depending on the type of event to be modelled. As observed in Figure 7b, the freeboard exceedance is not always lower than the water level outside the structure for the events obtained with steep waves.

To analyze the effects of the incident waves on green water evolution, some parameters were considered to define the duration of the events and characterize the freeboard exceedance, as shown in Figure 8. This figure shows the freeboard exceedance time series

of the eight consecutive events of CD2. It is important to mention that in most cases, the freeboard exceedance given by probe *d2* showed SVWS.

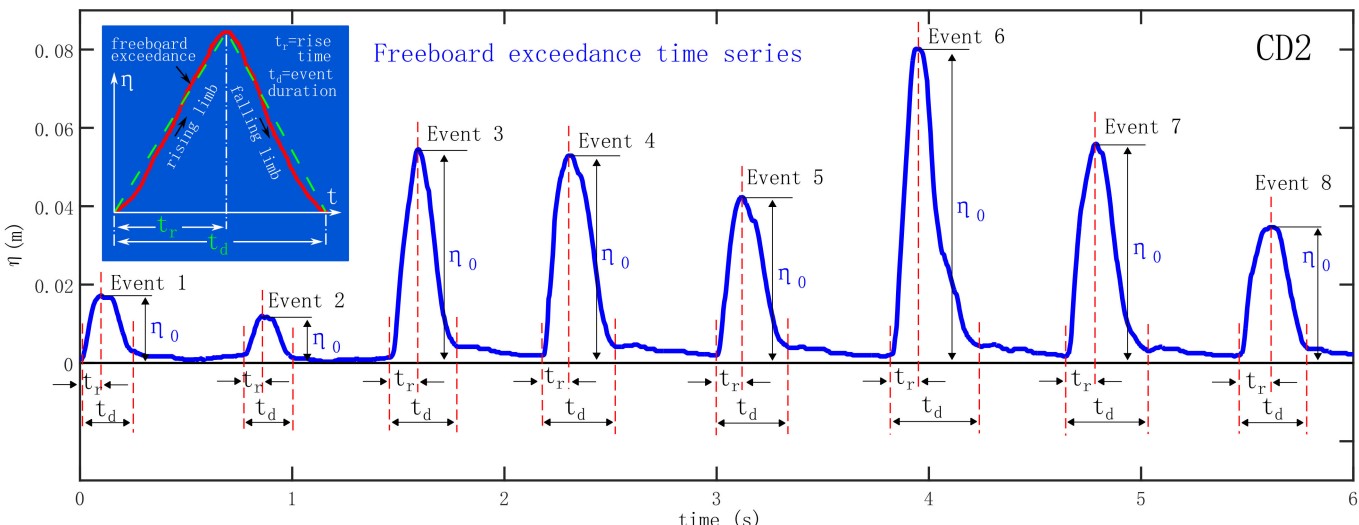

**Figure 8.** Time series of freeboard exceedance measured by probe *d2* from the deck level at the start of the structure, for the eight green water events observed in CD2.

In Figure 8, $\eta_0$ represents the maximum freeboard exceedance of each event, which corresponds to the peak value found between the rising and falling limbs of the signals. In a similar way as shown by [44–46] to characterize the temporal evolution of overtopping events in coastal structures, the rise time ($t_r$) and the event duration time ($t_d$) have been defined, as illustrated in the figure. In irregular sea conditions, for applications in coastal structures, Formentin and Zanuttigh [46] found that the ratio $t_r/t_d$ tends to increase as the overtopping flow evolves across the structure, staying mostly between 0.15–0.35 and not exceeding 0.5 [44]. The parameters shown in Figure 8 were obtained for all the study case events, to verify the effect of wave steepness on them.

Figure 9 shows the effect of wave steepness on the nondimensional maximum freeboard exceedance ($\eta_0/f$, where $f$ is the initial freeboard, 0.05 m). The data were differentiated according to the type of green water, as classified by [12], and wave height ($H_{w1}$–$H_{w3}$, see Table 1). The greatest freeboard exceedance was attained for PDB and HF types, at a steepness of 0.6–1.2, for $H_{w2}$ and $H_{w3}$. For the cases generated with $H_{w1}$ and $H_{w2}$, the greatest values were for HF-types. As expected, lower freeboard exceedances were found for waves with $H_{w1}$.

Figure 10 shows the effect of wave steepness on the ratio $t_r/t_d$. Note that in the figure, higher ratios were for waves with lower heights ($H_{w1}$ and $H_{w2}$) at lesser steepness, particularly for the events of DB and PDB types, whereas the lower $t_r/t_d$ ratios were found for $H_{w1}$ in PDB types. Most of the events with the highest waves ($H_{w3}$) were between $0.3 < t_r/t_d < 0.5$. The HF types of green water remained between this range for all cases. Note that higher $t_r/t_d$ ratios mean that the events took longer to attain the maximum freeboard exceedance ($\eta_0$), with respect to the duration of the event, whereas lower ratios mean that the maximum elevation was reached faster. It can thus be inferred that in some DB and PDB events, $\eta_0$ was reached more slowly than in HF events. $\eta_0$ was reached faster for some PDB types obtained with $H_{w2}$ at steepness of 0.08.

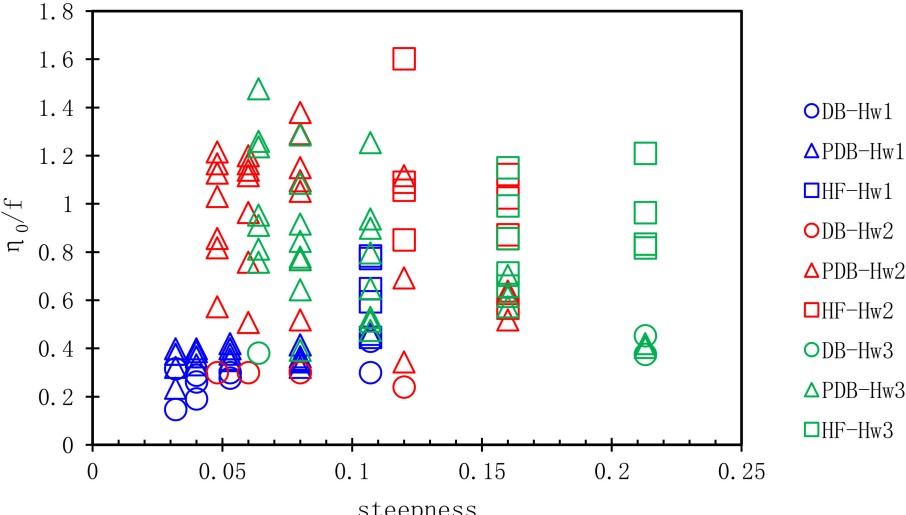

**Figure 9.** Effect of wave steepness ($\varepsilon = H_w/L_w$) on the nondimensional maximum freeboard exceedance ($\eta_0/f$, where $f$ is the initial freeboard, 0.05 m) for all the events of the 15 study cases.

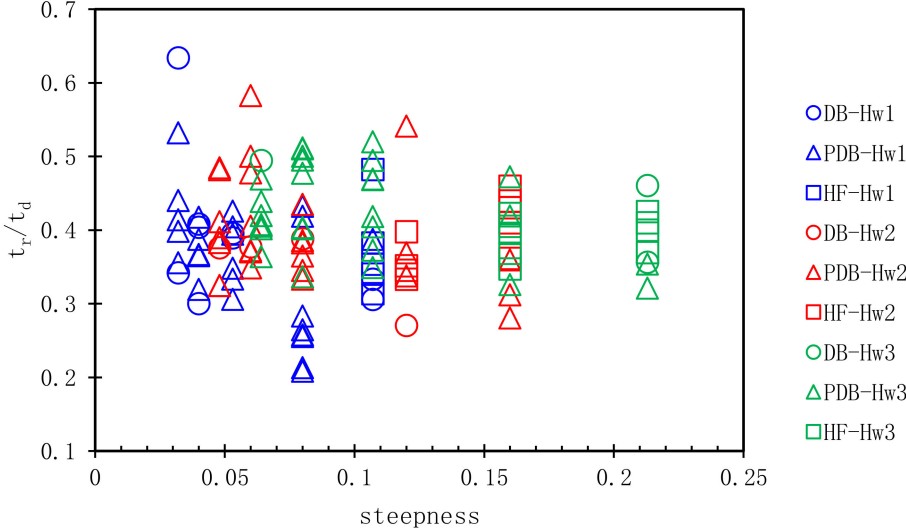

**Figure 10.** Effect of wave steepness ($\varepsilon = H_w/L_w$) on the $t_r/t_d$ ratio ($t_r$ and $t_d$ are rise and event duration times, respectively) for all the events of the 15 study cases.

The relation between the $t_r/t_d$ ratio and the maximum freeboard exceedance is shown in Figure 11. Maximum freeboard exceedances for all the wave heights are between $0.34 < t_r/t_d < 0.46$ and occurred for PDB and HF types of green water. The events with the shortest rise time ($t_r$) in the event duration ($t_d$) were in PDB types for $H_{w1}$, whereas the longest rise time was presented for $H_{w1}$, but in a DB type event. Overall, these results suggest that in most green water events, the time taken to reach the maximum freeboard exceedance did not exceed half of the duration of the event. This is in accordance with the threshold reported in [44] for coastal structures in irregular waves.

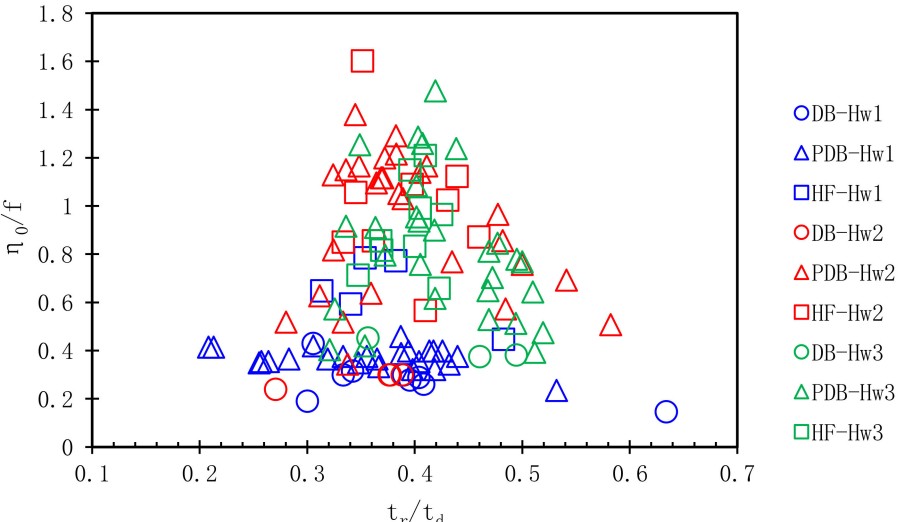

**Figure 11.** Relation between the nondimensional maximum freeboard exceedance ($\eta_0/f$, where $f$ is the initial freeboard, 0.05 m) and the $t_r/t_d$ ratio (where $t_r$ and $t_d$ are rise and event duration times, respectively) for all the events of the 15 study cases.

## 4. Green Water Volumes

As described in Section 2.2.2, the time series of total green water volumes ($\nabla$) over the deck length, visible in the images ($L = 0.3$ m), were obtained for all the study cases, and shown in non-dimensional form in Figure 12. These non-dimensional volumes ($\nabla^* = \nabla/L^3$, where $L$ is the deck length, 0.3 m) can be considered as non-dimensional vertical hydrostatic loads (see Section 2.2.2).

It can be seen in Figure 12a–e that for all the cases generated with $H_{w1}$ the lowest total volumes over the deck were found. In fact, this trend seems to be cumulative, probably due to the water layer remaining over the deck and the low freeboard exceedance of the events (Figure 9). Conversely, the highest volumes are shown in Figure 12a (CA2 and CA3), 12b (CB2 and CB3), and 12c (CC2), which were generally DB and PDB-types of green water [12], whereas the cases with more HF-types (Figure 12d,e) showed lower volumes over the deck. Note that these events were also shorter than the cases produced with longer waves.

An analysis similar to that in Section 3, was carried out to see the effect of wave steepness on the maximum green water volumes for each event in all the cases. Figure 13 shows this effect, differentiating between the types of green water events as reported in [12]. As expected, the $H_{w1}$ waves had lower values, whereas the highest waves ($H_{w2}$ and $H_{w3}$) gave the greatest values. Several of the events with PDB features showed the maximum volumes over the deck. In other words, higher volumes were found with waves with a steepness of 0.05–0.08. Maximum volumes in this range were more than twice the volumes obtained with steeper waves.

Figure 14 shows the relation between the non-dimensional maximum freeboard exceedance and the non-dimensional maximum green water volume for each event. The lowest volumes were attained for all the events obtained with $H_{w1}$, including all the DB-types of the experiments. Although the HF-types had considerable maximum freeboard exceedance values, almost of the order of the initial freeboard, the maximum non-dimensional volumes observed were mostly between 0.02 and 0.04. Although the maximum freeboard exceedance was for an HF-type event with $H_{w2}$, it did not necessarily cause the maximum global loads. On the other hand, the maximum volumes were obtained with PDB-type events with $H_{w2}$ and $H_{w3}$. It can be concluded that these events are the most common and are those with the most significant hydrostatic loads for the deck type considered in this study, when there is no backflow loading.

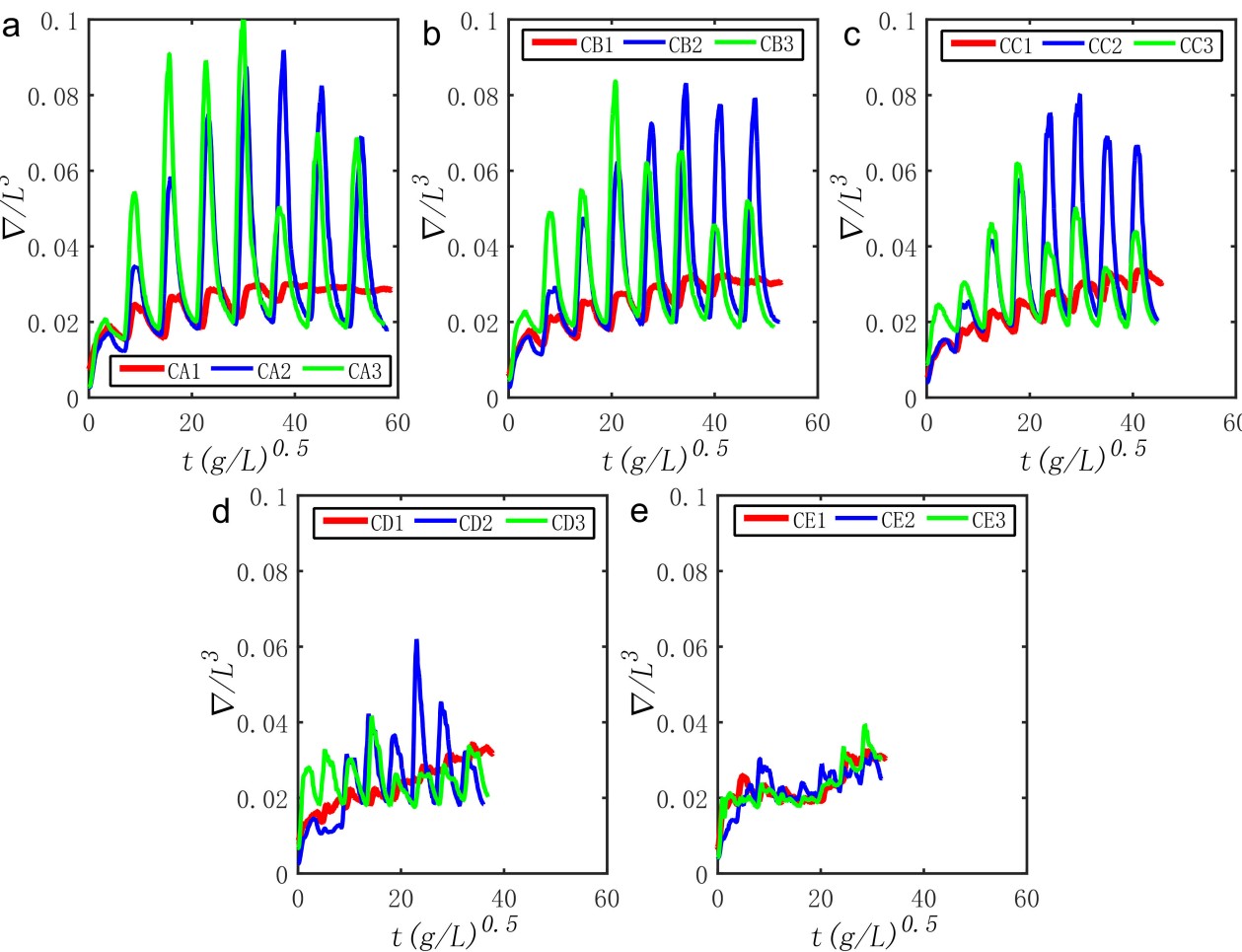

**Figure 12.** Time series of non-dimensional water volumes over the deck ($\nabla^* = \nabla/L^3$) for all the study cases. (**a**) Comparison between CA1, CA2, and CA3. (**b**) Comparison between CB1, CB2, and CB3. (**c**) Comparison between CC1, CC2, and CC3. (**d**) Comparison between CD1, CD2, and CD3. (**e**) Comparison between CE1, CE2, and CE3. $L = 0.3$ m, $g = 9.81$ m/s$^2$.

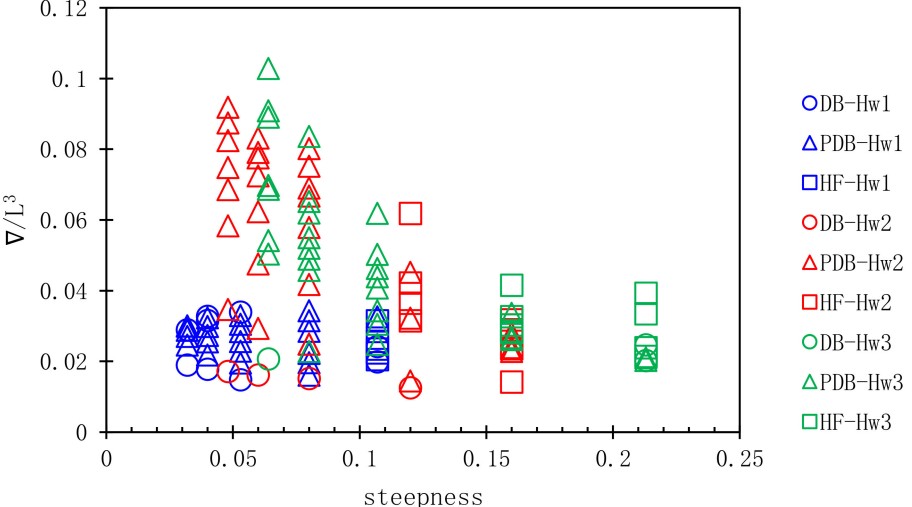

**Figure 13.** Effect of wave steepness ($\varepsilon = H_w/L_w$) in the non-dimensional green water volumes ($\nabla^* = \nabla/L^3$).

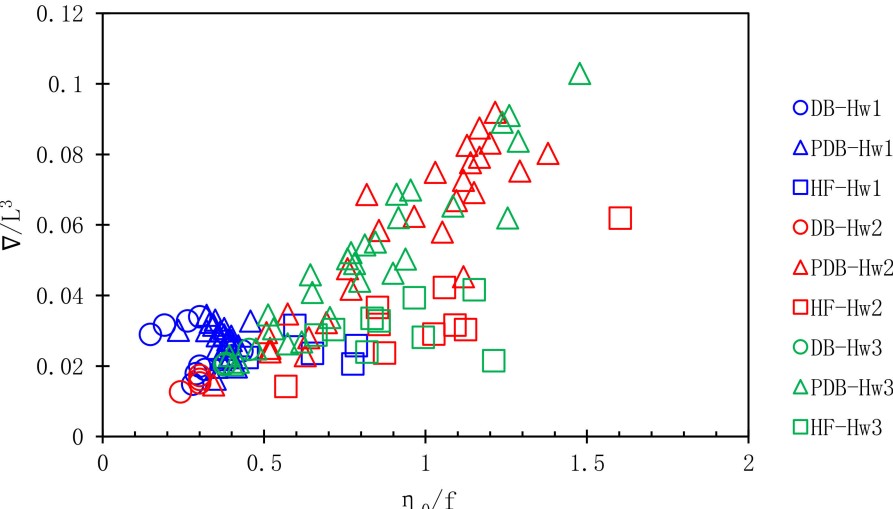

**Figure 14.** Relation between the non-dimensional freeboard exceedance ($\eta_0/f$) and the non-dimensional maximum volumes ($\nabla * = \nabla/L^3$) in the green water events of all the study cases.

## 5. On the Challenges to Assess Green Water Evolution

### 5.1. Factors That Influence Green Water Propagation

Assessing the propagation of water shipped onto the deck of a marine structure presents several challenges because such a phenomenon depends on several factors [1,2,8,9,13]. These are mainly the type of structure, the features of the incident flow, the operational conditions, and environmental interaction with another physical phenomenon, as illustrated in Figure 15.

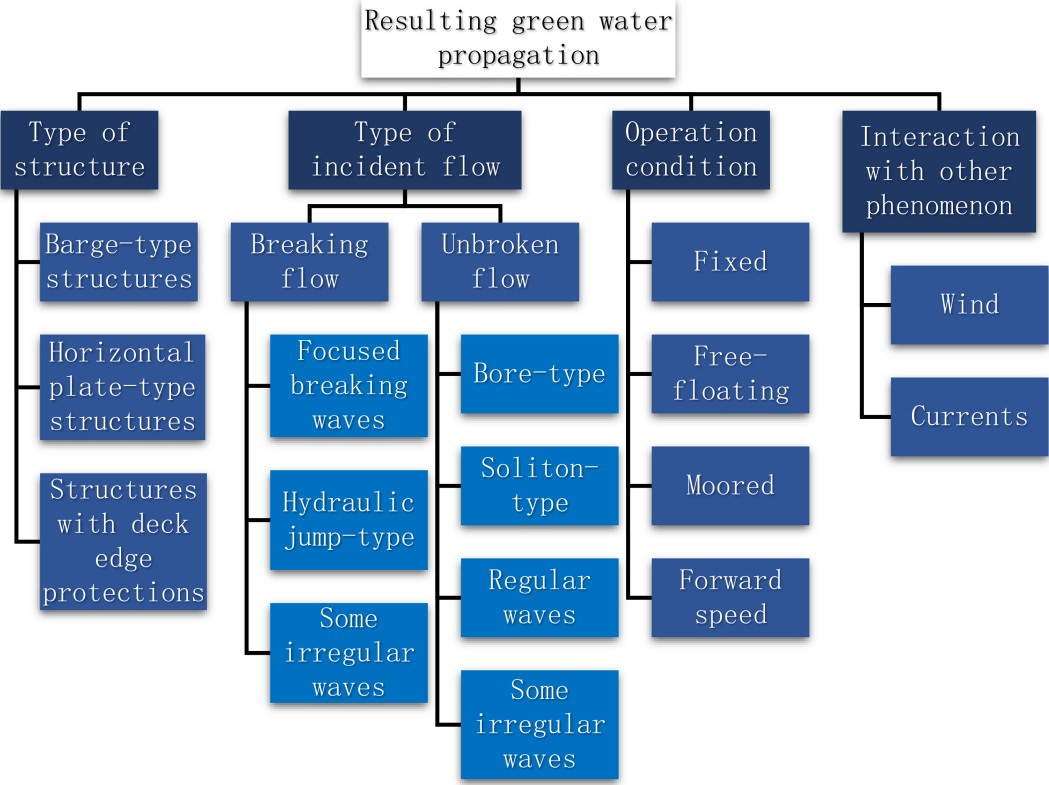

**Figure 15.** Factors that can influence green water propagation.

Type of structure: Several studies concern barge-type structures, using models that resemble rectangular boxes [9,11,17,47–49], as is the case of the present work. However, the

propagation of green water also matters in models that resemble platforms, such as those consisting of horizontal decks (e.g., [50,51]). The features of green water on a deck can drastically change when the structures have structural protections, such as bulwark plating at the start of the deck, as recently shown by [37]. Other types of protections on the deck of marine structures, such as those reported by [52,53], may also change the propagation of green water events.

Operation condition: Green water events can occur in structures operating in various conditions, which can have significant effects on the propagation of the water on deck. Research performed on fixed structures, for example [2,6,9,11,19,30,50,54,55], has been valuable in understanding important physical phenomena related to the patterns, elevations, and kinematics of such events. However, green water behavior can be different if the structures are moving, such as when they are floating freely (e.g., free heave-pitch oscillation [56]), when they are moored (e.g., [1,21,22,57]), or when they move with forward speed (e.g., [58–64]).

Type of incident flow: It has been reported in the literature [6–10] that the propagation of water over the deck differs considerably if the incident flow has unbroken or breaking features. With unbroken waves, it is possible to obtain types of green water such as those reported in [9,13,17], which resemble the DB, PDB, and HF types of this work. Some unbroken-type waves can be solitons, wet dam-break bores (e.g., [19]), as well as regular (e.g., [9,12,17]) and some irregular waves. On the other hand, when the waves have breaking features, violent interactions with the structures can be observed as suggested by [9,13], with unsteady flow in the form of splash followed by a compact mass of water (see, for instance the snapshots shown in [7,8,48,49]). These green water features can be found with focusing waves (e.g., [6,7]), incident flows resembling hydraulic jumps (e.g., [8]), and perhaps, some incident waves in irregular sea conditions that break before interacting with the structure.

Interaction with other phenomena: The propagation of green water over the deck may also be affected by the interaction of the incident waves with environmental phenomenon, such as wind and currents. Although such interaction is possible, and studies regarding green water in irregular sea conditions exist (e.g., [20,65,66]), to the authors knowledge, there are no detailed studies regarding the combined effect of wind-current-waves on green water evolution in marine structures. Perhaps this could be considered in further research.

### 5.2. Some Challenges in Assessing Green Water Propagation

In real conditions, green water is a complex, non-linear problem, in which 3D effects should be considered [1,2,58,59]. Although 3D research is valuable in evaluating realistic scenarios of this phenomenon, results from simplified 2D experiments, such as those presented in [2,9] provide useful information for modelling green water with analytical and numerical tools (e.g., [16,18]). Based on the experimental findings of the authors and results published elsewhere, four main challenges for the representation of green water evolution by analytical and numerical tools can be identified (Figure 16).

First, not all green water events start loading the deck as the flow crosses the deck edge. For design purposes, particularly when vertical loads matter, it can be convenient to consider the effective time of the interaction with the deck. This has been noted numerically by [16] and experimentally by [12]. Recently, Hernández-Fontes et al. [12] presented a simplified approach to consider "apparent" and "effective" interaction parameters.

Second, as discussed in this work, the evolution of some types of green water events presents MVWS elevations, especially at the start of the deck. Most comparisons between experiments and analytical or numerical results consider SVWS elevation measurements, as in [1,16,17,30,67,68]. Perhaps, in further comparisons regarding the initial stages of some types of events, such as PBD and HF-types, the use of multiple-valued water surfaces measurements would allow more detailed validations. This type of measurement may also be of importance in green water cases when there are protective structures on the deck (e.g., [37,52,53,69]).

Third, it is known that several types of green water events can entrap volumes of air during the initial stage of shipping in rectangular-type structures (e.g., PDB-types [2,9,11,13]). The analysis of cavity entrapment in the interaction of incident flows with vertical structures (i.e., "flip-through" phenomenon) has been reported in the literature [70–72]. For the case of green water, although relevant strategies have been proposed to investigate this type of phenomena [14] and detailed experiments have been reported [9–12], more research is still needed to simulate cavity evolution and to verify its effects on structural loading [37].

Finally, as described by [2,8], the flow patterns over the deck will depend on the features of the incident waves, which can be unbroken (e.g., [9–11,17]) or broken (e.g., [5–8,73,74]). In the latter, if unsteady and broken flow can be observed on the deck (see the snapshots in Figure 16 and in [4]), then the application of numerical models, rather than analytical models, could be used for detailed research. Although mesh-based Computational Fluid Dynamics (CFD) approaches are popular tools for fluid–structure interaction simulations, including green water (e.g., [38,57,75–79]), progress in CFD methods based on particles (e.g., [37,51,80–83]) could be a useful alternative, due to their ability to simulate large deformations and breaking flows [84]. A comprehensive review of these methods can be found in [85].

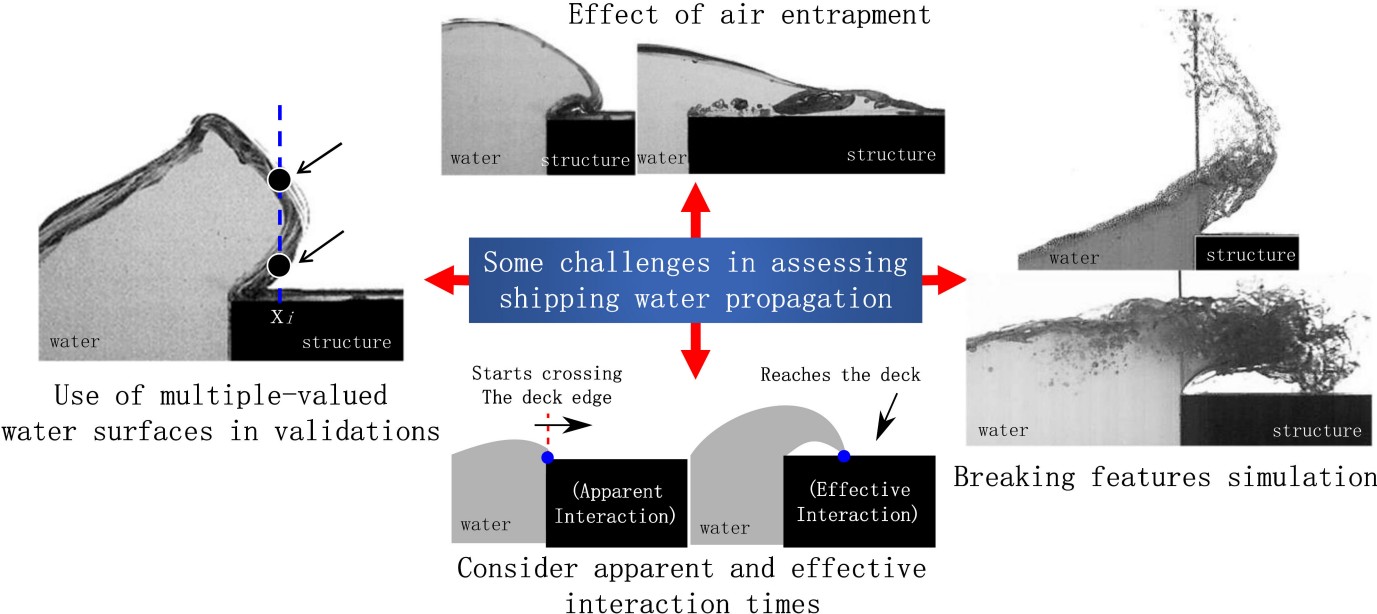

**Figure 16.** Some challenges in assessing green water propagation. Upper and left pictures are modified from the open-source database provided by [12]. The pictures on the right are modified from [4].

## 6. Conclusions

The experimental work presented by [12] was extended, to understand the evolution, in terms of water surface elevations and volumes over the deck, of different types of green water events, generated by wave trains, in a wave flume. The main results of the research can be listed as follows:

- Details in the generation of PDB and HF types of green water were examined. It was seen that they present MVWS elevations to be considered in comparisons of analytical and numerical models.
- Although all the events of this research are in the range of $0.18 < t_r/t_d < 0.64$, where $t_r$ and $t_d$ are the rise time to attain the maximum freeboard exceedance ($\eta_0$) and the event duration, respectively, $\eta_0$ occurred between $0.34 < t_r/t_d < 0.46$, for PDB and HF-types of green water.

- When $\eta_0$ is estimated from knowledge of the incident waves, it should be considered that the elevations are often lower than the wave elevation close to the structure. However, in HF-type events, the $\eta_0$ elevations could be higher.
- Although some HF events showed the maximum $\eta_0$ values in this research, the maximum volumes over the deck were observed for events resembling the PDB type of green water. Perhaps the HF-type could have relevance in impulsive loading at the beginning of the deck; this should be further explored.
- Since the green water features mainly depend on the type of structure, type of incident flow, operation conditions of the structure, and interaction with other physical phenomena, there is still a lot to do in performing detailed assessments of shipping water evolution for different types of events. For example, some challenges relate to the use of MVWS elevation measurements for a better comparison with model results. It is also important to consider the apparent and effective interaction times during the formation of the events, to attain more detailed breaking feature simulations, perhaps by meshless methods, and to increase research related to cavity entrapment phenomena, including its effects in the loading caused by flow on structures.
- It is hoped that the results presented here can contribute to more detailed comparisons of models, regarding the features in the initial stages of the generation of some types of green water events.

**Author Contributions:** Conceptualization: J.V.H.F.; methodology: J.V.H.F., I.D.H., E.M., and R.S.; software: I.D.H. and J.V.H.F.; validation: J.V.H.F. and I.D.H.; formal analysis: J.V.H.F., M.R.d.S., E.B.d.S., R.S.F.K., J.G., and L.T.; investigation: J.V.H.F., M.R.d.S., E.B.d.S., R.S.F.K., J.G., R.S., E.M., and P.T.T.E.; resources: E.M. and R.S.; data curation: J.V.H.F., I.D.H., M.R.d.S., E.B.d.S., R.S.F.K., J.G., and L.T.; writing—original draft preparation: J.V.H.F., E.M., and R.S.; writing—review and editing: J.V.H.F., E.M., R.S., and P.T.T.E.; visualization: J.V.H.F.; supervision: J.V.H.F., E.M., R.S., and P.T.T.E.; project administration: J.V.H.F.; funding acquisition: E.M. and R.S. All authors have read and agreed to the published version of the manuscript.

**Funding:** This research was funded by CONACYT-SENER-Sustentabilidad Energética, CEMIE-Océano project, Grant Agreement No. FSE-2014-06-249795.

**Institutional Review Board Statement:** Not applicable.

**Informed Consent Statement:** Not applicable.

**Data Availability Statement:** Not applicable.

**Acknowledgments:** E.B.d.S.: M.R.d.S, J.G., and R.S.F.K. thank the support provided by "Governo do Estado do Amazonas" with resources of "Programa de Apoio à Iniciação Científica da Fundação de Amparo à Pesquisa do Estado do Amazonas (PAIC/FAPEAM)" through call No. 045/2020-GR/UEA and UEA-SISPROJ projects 26893, 26898, 26899 and 26910, respectively. P.T.T.E. thanks the support provided by the Brazilian National Council for Scientific and Technological Development (CNPq). R.S. and E.M., thank the support provided by CONACYT-SENER-Sustentabilidad Energética, CEMIE-Océano project, Grant Agreement No. FSE-2014-06- 249795. The help provided by Jill Taylor for the revision of the manuscript is gratefully acknowledged.

**Conflicts of Interest:** The authors declare no conflict of interest.

## Appendix A

Error analysis. This appendix includes an analysis of the possible errors that can arise during the generation of binary images and subsequent image analysis, using the method of [29] to measure the multiple-valued water surface elevations. In that approach, the user can manually select a lower and an upper threshold in the image processing plug-in to binarize the images (see Figure 3). Keeping the upper level as constant (i.e., 255, as used in this work), the effect of different lower threshold levels (LTL) on the final water surface measurements has been evaluated to quantify a possible error. For the time series of the first green water event in CE3, measured by probe *d2*, Figure A1a shows the maximum root mean squared value (RMSE, see [29] for the formula) of measurements taken with different

LTLs with respect to the LTL considered in this work (i.e., LTL = 145). The snapshots below the graph illustrate the luminance effect to differentiate the flow from the background (i.e., different LTL values). Overall, maximum RMSE values are between 0.3 mm and 1.5 mm. Figure A1b shows the aspect of the measured time series, including upper and lower limits defined by the mean RMSE from the analysis done ($\pm$0.92 mm).

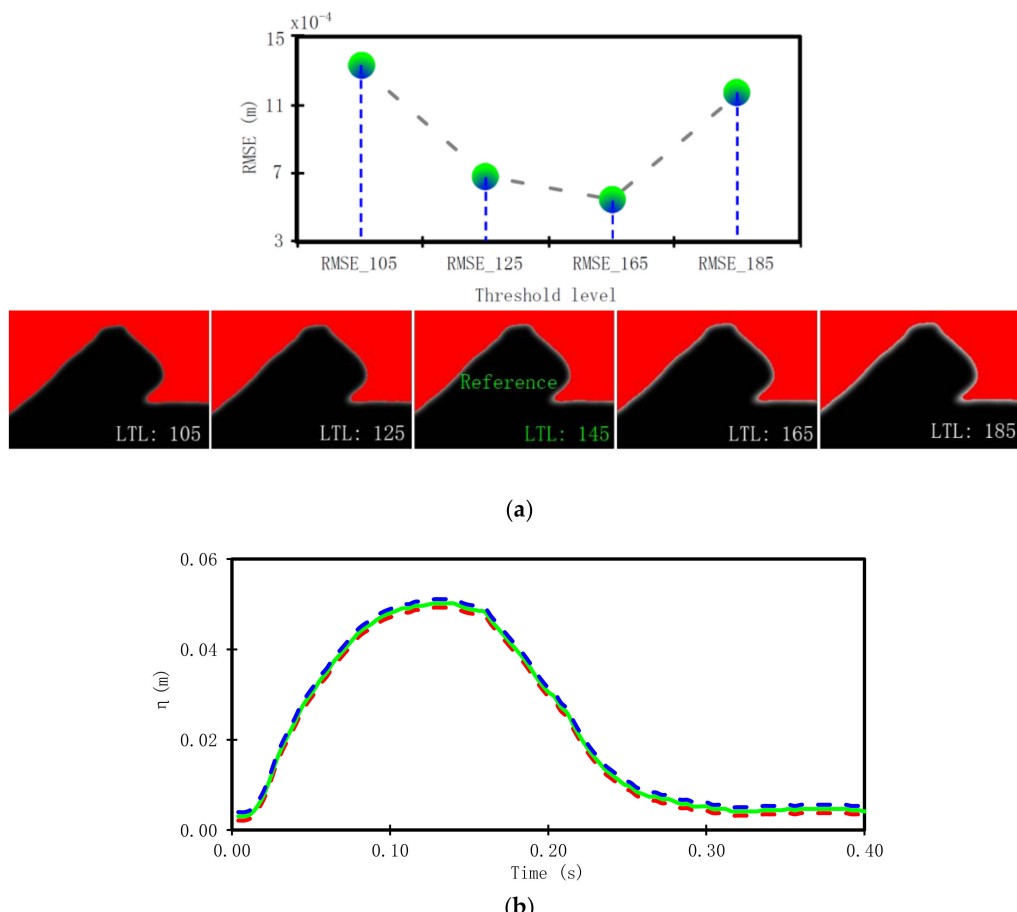

(a)

(b)

**Figure A1.** Possible error in measurements due to image binarization. (**a**) RMSE obtained for different threshold levels (LTL) before the binarization of the images. (**b**) Time series of water elevations by probe *d2* in the first event of CE3. The solid line represents the measurements with LTL = 145. The dashed lines denote the range of possible error (mean RMSE) because of different LTL values in the image processing.

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
