# Peer review of "On the Evolution of Different Types of Green Water Events"

_water, doi:10.3390/w13091148_

Round 1
Reviewer 1 Report
This manuscript by Hernández-Fontes et al. clearly progresses understanding into the evolution of green water events as they interact with a structure’s deck or platform surface. The content is well structured and concise, with high quality figures that are all relevant to the results presented. The introduction provides a good description of previous work. The target audience is very specific, I have suggested making a couple of generic statements in the introduction and conclusion to widen the interest a little. I have a few minor comments to improve the clarity of the manuscript as detailed below:
I would avoid the use of acronyms in the abstract. It should stand-alone and be understandable by a general audience.
P1 L28 “were investigated”
P1 L36 check the gramma “structures and shipping, propagating over”.
I’m not an expert in this area of research but there is similarities to my research in white water wave overtopping of coastal defences. I find it unusual to call these events green water events rather than green water overtopping events when first described. I assume the authors are expertise and have selected the correct terminology.
The opening to the introduction could have a couple of generic sentences (directly after the first sentence) to pose the issue for a wider audience. What are the damage implications/costs for green water events? Why is this research so important? A couple of examples could be provided. The conclusions would also benefit from a generic statement about how this research could be used/applied to improve structure design. In the results, it is clarified that the aim is to generate a data set to development of new models to simulate green water impact that can then be used in engineering. This motivation needs to be in the introduction and a concluding remark about the capability of the data to make progress in model developments is required.
Table 1, why were only wave parameters modified and not the still water level? Does the size of the freeboard have no influence on the green water evolution?
I’m not an expert in image processing, but the method seems logical and obtains the data required to advance the process understanding of green water wave evolution.
I am a little unclear if the freeboard exceedance is measured to the deck of the structure or to the mean water level adjacent to the structure, I believe it is the deck (Fig. 10), but to help clarify this it would be of value to add that to the caption in Fig. 11 and define this in the text. In Fig. 10, I would have thought the initial freeboard was the (positive) distance between the still water level and the deck, not the still water level as indicated by the arrow. I suggest editing the arrow so it is double headed to show it is the distance.
Freeboard exceedance is used before it is defined L262. Ensure definitions are given on first use. L261-264 provide a good introductory description of freeboard exceedance. This information could be provided in the introductory section, as it is the key parameter of interest in this study. The need to understand this parameter to develop numerical models is the background motivation for the research. Again I would suggest moving this to the introduction section.
L280/281. I’m not sure what is meant by “since the freeboard exceedance is not always lower than the water level outside the structure.”
Figs 12-17. The overlap in symbols makes it hard to see the different shapes. Could the symbols on top be a think outline rather than a filled shape or could the squares always be the bottom layer so the other shapes still allow their larger outline to be seen.
L472. Remove the space in “o f”.
Reviewer 2 Report
Link to novelty is weak. Lot of literature about presented issue have been done. Authors do not explain well, where is the novelty of the distingusihed method. Conclusion is not sufficiently described. It is more like summary of information, what can be read in previous chapters.
The manuscript is beyond the scope of the journal.
The manuscript entitled: Journal of Marine Science and Engineering would be more suitable.
Reviewer 3 Report
This paper describes some detailed measurements of long-crested waves impinging on a fixed structure. Optical methods are use to get details of green water over the deck. The discussion of previous work in the field and remaining challenges is particularly good.
Clarification is need for some of the results that puzzled me. The waves were supposed to be regular, but the incident crest heights in Figure 10 vary greatly for consecutive waves. It seems like plotting the variation of green water statistics as a function of incident crest height could remove a lot of the scatter in Figures 12-14. Why is there so much difference in the incident crest heights as the waves approach the structure in Figure 10b?
I printed out the paper to read it, but had to go back to the electronic version to be able to read some of the figures. The Figures will be even smaller in the published version so something should be done about this problem. Increasing the size of the legend in Figure 10 would help. The text in Figures 2 and 18 is almost unreadable. Taking the colored background off in Figure 18 would help but I'm not sure what could be done about Figure 2 except for gust getting rid of the pictures.
Round 2
Reviewer 2 Report
In the introduction, lack of connection of the state of the art to the paper goals. Currently, this is not performed in a convincing way. Please follow the literature review with a clear and concise state of the art analysis. This should clearly show the knowledge gaps identified and link them to your paper goals. Please reason both the novelty and the relevance of your paper goals. The original developments have to be properly described and reasoned. In reference section abbreviation of journal papers and doi number should be presented.
Round 3
Reviewer 2 Report
I accept in the present form
the manuscript entitled: On the evolution of different types of green water events.
Best regards